# Cilostazol Stimulates Angiogenesis and Accelerates Fracture Healing in Aged Male and Female Mice by Increasing the Expression of PI3K and RUNX2

**DOI:** 10.3390/ijms25020755

**Published:** 2024-01-06

**Authors:** Maximilian M. Menger, Maximilian Emmerich, Claudia Scheuer, Sandra Hans, Sabrina Ehnert, Andreas K. Nüssler, Steven C. Herath, Konrad Steinestel, Michael D. Menger, Tina Histing, Matthias W. Laschke

**Affiliations:** 1Department of Trauma and Reconstructive Surgery, BG Trauma Center Tuebingen, Eberhard Karls University Tuebingen, 72076 Tuebingen, Germany; 2Institute for Clinical and Experimental Surgery, Saarland University, 66421 Homburg, Germany; 3Department of Trauma and Reconstructive Surgery, BG Trauma Center Tuebingen, Siegfried Weller Institute for Trauma Research, Eberhard Karls University Tuebingen, 72076 Tuebingen, Germany; 4Institute of Pathology and Molecular Pathology, Bundeswehrkrankenhaus Ulm, 89081 Ulm, Germany

**Keywords:** mice, cilostazol, aging, fracture healing, angiogenesis, RUNX2, PI3K, femur, screw

## Abstract

Fracture healing in the aged is associated with a reduced healing capacity, which often results in delayed healing or non-union formation. Many factors may contribute to this deterioration of bone regeneration, including a reduced ‘angiogenic trauma response’. The phosphodiesterase-3 (PDE-3) inhibitor cilostazol has been shown to exert pro-angiogenic and pro-osteogenic effects in preclinical studies. Therefore, we herein analyzed in a stable closed femoral fracture model whether this compound also promotes fracture healing in aged mice. Forty-two aged CD-1 mice (age: 16–18 months) were daily treated with 30 mg/kg body weight cilostazol (*n* = 21) or vehicle (control, *n* = 21) by oral gavage. At 2 and 5 weeks after fracture, the femora were analyzed by X-ray, biomechanics, micro-computed tomography (µCT), histology, immunohistochemistry, and Western blotting. These analyses revealed a significantly increased bending stiffness at 2 weeks (2.2 ± 0.4 vs. 4.3 ± 0.7 N/mm) and an enhanced bone formation at 5 weeks (4.4 ± 0.7 vs. 9.1 ± 0.7 mm^3^) in cilostazol-treated mice when compared to controls. This was associated with a higher number of newly formed CD31-positive microvessels (3.3 ± 0.9 vs. 5.5 ± 0.7 microvessels/HPF) as well as an elevated expression of phosphoinositide-3-kinase (PI3K) (3.6 ± 0.8 vs. 17.4 ± 5.5-pixel intensity × 10^4^) and runt-related transcription factor (RUNX)2 (6.4 ± 1.2 vs. 18.2 ± 2.7-pixel intensity × 10^4^) within the callus tissue. These findings indicate that cilostazol accelerates fracture healing in aged mice by stimulating angiogenesis and the expression of PI3K and RUNX2. Hence, cilostazol may represent a promising compound to promote bone regeneration in geriatric patients.

## 1. Introduction

The elderly population, commonly defined as those aged 65 and older, is expected to grow rapidly in the coming decades, which makes the treatment of geriatric patients in trauma and orthopedic surgery a major health issue [1]. Aged individuals not only suffer from a higher risk of bone fractures and mortality but also exhibit a reduced capacity for bone regeneration [2,3]. This may result in delayed bone healing and non-union formation, which is associated with a significant loss of function for the patients as well as extensive revision surgeries [4]. Multiple factors may contribute to the age-related deterioration of fracture healing, such as a decreased number and worsened function of stem cells, structural changes in the periosteum, and impaired angiogenesis [5]. In fact, affected fracture healing in the aged is markedly associated with age-related dysfunction of the bone vascular system, resulting in delayed vascularization at the fracture site [5].

Cilostazol, a phosphodiesterase-3 (PDE-3) inhibitor, induces blood vessel dilatation and hampers platelet aggregation by elevating cyclic adenosine monophosphate (cAMP) levels [6]. Preclinical studies further demonstrated that cilostazol increases tissue perfusion and promotes angiogenesis [7,8]. Moreover, PDE inhibitors are capable of stimulating the differentiation of osteoblasts [9,10]. In line with these findings, we recently showed that cilostazol accelerates fracture healing in young adult mice by upregulating the expression of the pro-angiogenic cysteine-rich angiogenic inducer (CYR)61 [11]. In contrast, the effects of cilostazol on fracture healing in the aged have not been analyzed so far. However, this investigation is essential, considering the fact that aging results in an impaired functional reserve of several organs. Accordingly, pharmacokinetics, metabolism, and drug effects may significantly differ in old adults when compared to young individuals [12].

Accordingly, in the present study, we analyzed for the first time the effects of cilostazol on fracture healing in aged mice by means of a stable closed femoral fracture model. For this purpose, the animals were either treated daily with cilostazol or vehicle for 2 or 5 weeks after fracture. Subsequently, the callus tissue was analyzed by radiological, biomechanical, histological, immunohistochemical, and protein biochemical techniques.

## 2. Results

The femora of aged mice were analyzed according to the experimental protocol described in detail in the Materials and Methods sections. Twenty-one mice received daily 30 mg/kg body weight cilostazol dissolved in 100 µL saline by oral gavage starting on the day of fracture. Control animals (*n* = 21) received equal amounts of vehicle (saline). At 2 weeks and 5 weeks after fracture, the femora were analyzed by means of biomechanics, µ-CT, histology, and immunohistochemistry. In addition, Western blot analysis for the quantification of protein expression was performed at 2 weeks after fracture (Figure 1).

### 2.1. X-ray

The X-rays taken at 2 weeks after fracture showed clear signs of callus formation in cilostazol-treated animals and controls (Figure 2a,c). At 5 weeks, the femora in both groups exhibited complete osseous bridging (Figure 2b,d). Of note, implant dislocation was not observed in any of the animals operated for the present study.

### 2.2. Biomechanics

Our biomechanical analysis revealed a significantly higher absolute and relative bending stiffness of the femora in cilostazol-treated mice at 2 weeks after fracture when compared to controls (Figure 2e,g). This indicates accelerated fracture healing in cilostazol-treated animals. At 5 weeks after fracture, the absolute and relative bending stiffness did not differ anymore between the two groups (Figure 2f,h). Notably, there was also no difference in the bending stiffness of unfractured femora between the two groups (100.6 ± 5.0 N/mm (control) vs. 92.3 ± 5.2 N/mm (cilostazol)).

### 2.3. µCT

In line with our radiographic analysis, µCT imaging also showed clear signs of endochondral bone healing with callus formation at 2 weeks and osseous bridging at 5 weeks after fracture in control animals (Figure 3a,b) and cilostazol-treated mice (Figure 3c,d). At 2 weeks after fracture, our µCT analysis showed no significant differences in bone volume between the two groups (Figure 3e). At 5 weeks, however, we found a significantly increased bone volume in cilostazol-treated mice when compared to controls (Figure 3f). Further analysis of the trabecular architecture demonstrated no significant difference in trabecular separation, thickness, and number at 2 weeks after fracture (Figure 3g and Table 1). In contrast, we detected a significantly lower trabecular separation in cilostazol-treated animals at 5 weeks (Figure 3h). Notably, the trabecular thickness and number did not significantly differ between the two study groups at 5 weeks after fracture (Table 1).

### 2.4. Histomorphometry and Histology

Our histomorphometric analysis showed clear signs of endochondral callus formation at 2 weeks and complete osseous bridging at 5 weeks after fracture in both groups (Figure 4a–d). The quantitative analysis of the callus composition at 2 weeks demonstrated a lower amount of fibrous tissue (27.5 ± 4.5%) and a higher amount of osseous tissue (42.2 ± 5.2%) in cilostazol-treated animals when compared to controls (45.4 ± 8.9% and 30.2 ± 5.7%, respectively). However, these differences did not prove to be statistically significant (Figure 4e). Moreover, the amount of cartilage tissue did not significantly differ between the two groups (Figure 4e). At 5 weeks after fracture, major parts of the callus consisted of osseous tissue in both groups, whereas cartilaginous tissue was nearly absent (Figure 4f). Overall, there was no significant difference in the callus composition between cilostazol-treated mice and controls at 5 weeks after fracture (Figure 4f). However, the evaluation of fracture bridging demonstrated a significantly higher histology score in cilostazol-treated mice at 2 weeks when compared to controls (Figure 4g). At 5 weeks after fracture, however, the histology score did not significantly differ between the two groups (Figure 4h).

### 2.5. Immunohistochemistry

We further investigated the vascularization of the callus tissue by immunohistochemistry. Interestingly, our analysis showed a significantly higher number of CD31-positive microvessels within the callus tissue of cilostazol-treated mice at 2 weeks after fracture when compared to controls (Figure 5a,c,e). At 5 weeks, the number of microvessels did not significantly differ between the two groups (Figure 5b,d,f).

### 2.6. Western Blot

Western blot analysis at 2 weeks after fracture revealed an over 2-fold higher expression of BMP-2 in cilostazol-treated mice when compared to controls (Figure 6a,b). However, the difference did not prove to be statistically significant. The analysis of BMP-4 (Figure 6a,c) and CYR61 (Figure 6d,e) expression within the callus tissue demonstrated no significant difference between the two groups. However, in line with our immunohistochemical data, we found a significantly increased expression of CD31 in cilostazol-treated mice (Figure 6d,f). Additional analyses also showed a significantly higher expression of PI3K and RUNX2 within the callus tissue of cilostazol-treated animals when compared to controls (Figure 6g–i).

## 3. Discussion

In the present study, we analyzed for the first time the effects of cilostazol on fracture healing in aged mice. Our novel data demonstrate that cilostazol treatment accelerates fracture healing and stimulates bone formation in these animals. This is associated with an increased bending stiffness of the femora, a higher number of CD31-positive microvessels, and enhanced expression of PI3K and RUNX2 within the callus tissue.

Many factors contribute to successful bone regeneration. Among them, angiogenesis is of pivotal importance. Newly formed blood vessels transport essential nutrients, oxygen, and stem cells to the fracture site, which enables the formation of a callus tissue and the subsequent process of ossification and remodeling [4]. Notably, aging is associated with an impairment of the bone vascular system and may therefore delay the vascularization of the callus tissue [5]. Accordingly, Lu et al. [13] found that the callus tissue of young animals contains a higher number of blood vessels when compared to middle-aged and geriatric mice. This was associated with a reduced expression of matrix metalloproteinase (MMP)-9 and -13 in aged animals during the early phase of fracture healing [13]. Previous studies reported that cilostazol is capable of stimulating angiogenesis in ischemic hindlimb models [8,14]. In line with this observation, we detected a significantly higher number of CD31-positive microvessels within the callus tissue of femora in cilostazol-treated mice at 2 weeks after fracture when compared to controls. Hence, we suggest that this improved early vascularization may have markedly contributed to accelerated fracture healing under cilostazol treatment.

PI3K signaling is activated through receptor tyrosine kinases and G-protein coupled receptors and regulates the proliferation and differentiation of osteoblasts during osteoblastogenesis and skeletal remodeling [15]. Moreover, Scanlon et al. [16] reported that a global increase in PI3K signaling results in improved femoral bone regeneration by stimulating the proliferation of periosteal cells during the early phase of fracture healing. Interestingly, there is evidence that cilostazol treatment also results in an increased PI3K expression. For instance, Mohamed et al. [17] demonstrated that cilostazol reduces diabetes-induced damage of testicular tissue by elevating PI3K expression. Hence, it may be assumed that cilostazol improves fracture healing in aged mice by PI3K-mediated stimulation of osteoblastogenesis. In addition, Shi et al. [18] showed that cilostazol suppresses apoptotic cell death and reduces dysfunction of endothelial cells via the activation of the PI3K/Akt pathway. This effect may mitigate the age-associated deterioration of vascularization at the fracture site, resulting in an increase in angiogenesis and ultimately in an acceleration of fracture healing.

RUNX2 is essential for osteoblast differentiation and chondrocyte maturation. It acts as a regulator of bone formation by controlling several signaling pathways, including hedgehog, FGF as well as Wnt, and, thus, stimulates the differentiation of mesenchymal stem cells to the osteogenic lineage [19]. Furthermore, Zelzer et al. [20] demonstrated that RUNX2 is essential for endochondral bone formation. In RUNX2-deficient mice, the invasion of cartilaginous tissue with newly formed blood vessels does not occur [20]. Moreover, RUNX2 cDNA-loaded type I collagen sponges have been shown to enhance bone tissue regeneration within alveolar defects in mice [21]. Because we also detected a higher RUNX2 expression within the callus tissue of fractured femora in cilostazol-treated mice, we assume that cilostazol simultaneously activates several pathways in aged mice that accelerate fracture healing and improve bone formation. This view is further supported by the fact that we also detected an over 2-fold higher expression of BMP-2 within the callus tissue under cilostazol treatment. BMP-2 plays a major role in the process of bone regeneration, particularly during the early stage of healing, by stimulating the differentiation of mesenchymal stem cells [22,23,24,25].

In contrast to our previous study, in which we analyzed the effects of cilostazol on fracture healing in young adult mice, we herein did not observe a significantly increased expression of the pro-osteogenic and pro-angiogenic marker CYR61 [11,26]. This indicates that the mode of action of cilostazol in aged mice markedly differs from that in young adult animals. In fact, these findings are in line with our previous studies analyzing the effects of pantoprazole on fracture healing in young adults and aged mice [27,28]. Our data indicate a different mode of action of pantoprazole treatment in aged mice when compared to young adult animals. In aged mice, pantoprazole causes impaired bone healing by stimulating an overwhelming osteoclastic response, most likely due to an enhanced parathyroid hormone (PTH) serum level by calcium malabsorption. On the other hand, in young adult animals, pantoprazole exerts impaired healing due to a reduction in osteoclast activity with delayed bone remodeling, which is hypothesized to be caused by an inhibition of the osteoclastic V-ATPase [28].

In conclusion, the present study demonstrates that cilostazol accelerates fracture healing and induces bone formation in aged CD-1 mice by stimulating angiogenesis and increasing the expression of PI3K and RUNX2 within the callus tissue. Hence, this compound, which is an approved and well-established drug in clinical practice, may represent a promising candidate to promote bone regeneration in geriatric patients.

## 4. Materials and Methods

### 4.1. Animals

For this study, 42 male and female CD-1 mice with an age of 16–18 months and a body weight of 35–45 g were used. The age of 16–18 months was chosen according to previous reports demonstrating age-associated physiological dysfunctions and tumor development in CD-1 mice [29]. Of note, male and female mice were evenly distributed between the two study groups, to avoid the influence of sex on fracture healing. All mice were housed under a 12/12-h day/night cycle and had free access to water and standard pellet chow (Altromin, Lage, Germany).

All animal experiments were approved by the local governmental animal protection committee (permit number: 04/2019). The study was executed according to the European legislation on the protection of animals (Directive 2010/63/EU) and the National Institutes of Health (NIH) Guide for the Care and Use of Laboratory Animals (Institute of Laboratory Animal Resources, National Research Council, Washington, DC, USA).

### 4.2. Cilostazol Treatment

Twenty-one mice received daily 30 mg/kg body weight cilostazol (Pletal^®^, Otsuka, Wexham, UK) dissolved in 100 µL saline by oral gavage starting on the day of fracture. Control animals (*n* = 21) received equal amounts of vehicle (saline). The cilostazol dosage was chosen according to our previous study in young adult mice [11]. The animals were sacrificed using an overdose of pentobarbital (Nacoren^®^, Boehringer Ingelheim Vetmedica GmbH, Ingelheim am Rhein, Germany) and their femora were harvested at 2 weeks (*n* = 8 in each group) and 5 weeks (*n* = 9 in each group) after fracture to investigate bone healing by biomechanical, radiological, histological, and immunohistochemical analyses. In addition, 8 mice (*n* = 4 in each group) were sacrificed for Western blot analyses at 2 weeks after fracture (Figure 1). 

### 4.3. Fracture Model

Closed femoral fractures were induced in CD-1 mice and stabilized using the MouseScrew (AO Development Institute, Davos, Switzerland), as described previously in detail [11,27,28].

All procedures were performed under aseptic conditions using an operating microscope to achieve adequate precision. The mice were anesthetized with an intraperitoneal (i.p.) injection of ketamine (75 mg/kg body weight; Ursotamin^®^, Serumwerke Bernburg, Bernburg, Germany) and xylazine (15 mg/kg body weight; Rompun^®^, Bayer, Leverkusen, Germany). A medial parapatellar incision of ~4 mm was created at the right knee and the patella was dislocated laterally. At the intercondylar notch, the intramedullary canal was opened using a trephine (diameter: 0.5 mm). Additionally, the greater trochanter was drilled retrogradely over the intramedullary cavity using an injection needle (diameter: 0.4 mm). Then, a tungsten guide wire (diameter: 0.2 mm) was inserted through the needle, which was removed thereafter. The femur was fractured using a 3-point bending device, as described previously [5]. Afterward, a MouseScrew was implanted through the intercondylar notch over the guide wire, resulting in stable osteosynthesis by interfragmentary compression. In the end, the guide wire was removed, the patella was repositioned, and the incision was closed with 5-0 synthetic sutures. The implant position was confirmed by radiography (MX-20, Faxitron X-ray Corporation, Wheeling, IL, USA). For analgesia, the mice received 5 mg/kg body weight carprofen (Rimadyl™, Zoetis GmbH, Berlin, Germany) subcutaneously on the day of surgery. Additionally, tramadol hydrochloride (Grünenthal, Aachen, Germany) was added to the drinking water (1 mg/mL) starting one day prior to surgery until three days after surgery.

### 4.4. X-ray

Before harvesting the femora, lateral radiographs were performed to check for implant or fracture dislocation at 2 and 5 weeks after fracture (MX-20, Faxitron X-ray Corporation, Wheeling, IL, USA).

### 4.5. Biomechanics

After the removal of the soft tissue, the bending stiffness of the harvested fractured femora was measured by a non-destructive approach using a 3-point bending device [11]. All femora were mounted to the bending device with the ventral aspect facing upwards to guarantee standardized measuring conditions. The fracture zone was placed directly under the middle punch with a working gauge length of 6 mm between the two bearing areas at the edges. Due to the different stages of bone healing, the applicable loads varied between the individual femora. Loading was stopped individually when the load-displacement curve deviated > 1% from linearity. The absolute bending stiffness (N/mm) was then calculated from the load-displacement diagram. The unfractured left femora were also analyzed to account for differences in bone stiffness of the individual animals. Accordingly, the relative bending stiffness was additionally assessed as a percent of the corresponding unfractured femora (%).

### 4.6. Micro-Computed Tomography (µCT)

The femora were scanned (Skyscan 1176, Bruker, Billerica, MA, USA) at a spatial resolution of 9 µm with a standardized setup (tube voltage: 50 kV; current: 200 µA; intervals: 0.4°; exposure time: 3500 ms; filter: 0.5 mm aluminum). Images were stored in three-dimensional arrays. To express gray values as mineral content, hydroxyapatite (CaHA) phantom rods with known bone mineral density (BMD) values were employed for calibration. Bone volume (mm^3^), trabecular separation (mm), trabecular thickness (mm), and trabecular number (1/mm) were calculated from the callus region of interest for each specimen.

### 4.7. Histology and Histomorphometry

For histological analyses, the bones were fixed in paraformaldehyde for 24 h. Subsequently, the specimens were embedded in a 30% sucrose solution for another 24 h and then frozen at −80 °C. Longitudinal sections through the femoral axis with a thickness of 4 µm were cut by the Kawamotos film method [30] and stained with safranin-O for histomorphometric analyses. At a magnification of 12.5× (Olympus BX60 Microscope, Olympus, Shinjuku, Japan; Zeiss Axio Cam and Axio Vision 3.1, Zeiss, Oberkochen, Germany) structural indices were calculated according to the recommendations of Gerstenfeld et al. [31]. The following histomorphometric parameters of the bone defects were evaluated: (i) total callus area; (ii) bone callus area; (iii) cartilaginous callus area; and (iv) fibrous callus area. The total callus area is defined as the total area of bone, cartilaginous, and fibrous callus area.. The cartilaginous callus is clearly indicated by the red color with safranin-O staining. The fibrous callus is characterized by long green fibers within the center of the callus tissue. Each area was marked and calculated using the ImageJ analysis system (NIH, Bethesda, MD, USA). Moreover, we used a scoring system (histology score) to evaluate the quality of fracture bridging. Both cortices were analyzed for bone bridging (2 points), cartilage bridging (1 point), or bridging with fibrous tissue (0 points). This scoring system results in a maximum of 4 points for each specimen, indicating complete bone bridging. Notably, areas with no visible tissue within the callus were excluded from the measurement.

### 4.8. Immunohistochemistry

To assess vascularization, additional longitudinal tissue sections (4 µm) were cut at 2 and 5 weeks after fracture. For immunohistochemistry, the sections were stored for 60 secs in 100% ethanol. After washing in phosphate-buffered saline (PBS), the sections were permeabilization in PBS with 0.2% Triton-X (PBST) for 10 min. The process was blocked by 5% goatserum in PBST for 1 h. For the immunohistochemical detection of microvessels, sections were stained with a monoclonal rat anti-mouse antibody against the endothelial cell marker CD31 (1:100; Abcam, Cambridge, UK) overnight in PBST with 5% goat serum. Subsequently, the sections were washed again in PBS, and stained with a goat anti-rat IgG-Alexa555 antibody as the secondary antibody (1:100; Life Technology, Eugene, OR, USA) in PBS for 1 h. Cell nuclei were stained with Hoechst 33342 (2 µg/mL; Sigma-Aldrich, Taufkirchen, Germany). Finally, the sections were covered with glycerin gelatine.

In 2-week specimens, one high-power field (HPF, ×400 magnification) was placed in a standardized manner in the central region of the callus (former fracture gap), while five additional HPFs were placed at each site within the periosteal region of the callus. Due to the reduced size of the callus at 5 weeks, only 3 additional HPFs were placed at each site within the periosteal region of the callus. The number of CD31-positive microvessels within each HPF was counted and the mean for each specimen was determined.

### 4.9. Western Blot

Protein expression within the callus tissue at 2 weeks after surgery (*n* = 4 each group) was determined by Western blot analysis, including the expression of bone morphogenetic protein-2 (BMP-2) and -4 (BMP-4), the osteogenic and angiogenic CYR61, endothelial cell marker CD31, intracellular signal transducer enzyme phosphoinositide-3-kinase (PI3K), and runt-related transcription factor (RUNX)2, an inducer of osteoblast and chondrocyte differentiation. The callus tissue was frozen and stored at −80 °C until required. After saving the whole protein fraction, the analysis was performed using the following monoclonal antibodies: goat anti-mouse BMP2 and BMP4 (1:300, R&D Systems, Wiesbaden, Germany), sheep anti-mouse CYR61 (1:300, R&D Systems), rabbit anti-mouse CD31 (1:300, Cell Signaling Technology, Danvers, MA, USA), mouse anti-mouse PI3K (1:100, Santa Cruz Biotechnology, Heidelberg, Germany), and rabbit anti-mouse RUNX2 (1:300, Abcam). Primary antibodies were followed by corresponding horseradish peroxidase-conjugated secondary antibodies (Dako Deutschland GmbH (Agilent), Hamburg, Germany). Protein expression was visualized by means of luminol-enhanced chemiluminescence after exposure of the membrane to the Intas ECL Chemocam Imager (Intas Science Imaging Instrument GmbH, Göttingen, Germany) and normalized to β-actin signals (1:1000, mouse anti-mouse β-actin, Santa Cruz Biotechnology) to correct for unequal loading. For quantification of protein expression, the LabImage 1D software (Version 4.1, Kapelan Bio-Imaging GmbH, Leipzig, Germany) was used. The software allows for the automatic detection of the position of Western blot bands and the quantification of pixel intensity.

### 4.10. Statistical Analysis

All data are given as means ± SEM. After testing the data for normal distribution (Kolmogorov–Smirnov test) and equal variance (*F*-test), comparisons between the two groups were performed using the unpaired Student’s *t*-test. For non-parametrical data, a Mann–Whitney U-test was used. All statistics were performed using the SigmaPlot 13.0 software (Jandel Corporation, San Rafael, CA, USA). A *p*-value of <0.05 was considered to indicate significant differences.

## Figures and Tables

**Figure 1 ijms-25-00755-f001:**
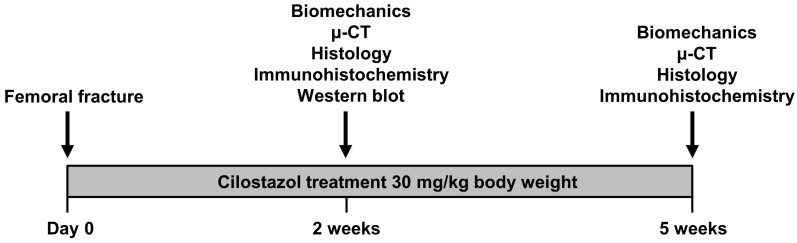
Experimental protocol of the cilostazol treatment. Twenty-one mice received daily 30 mg/kg body weight cilostazol dissolved in 100 µL saline by oral gavage starting on the day of fracture. Control animals (*n* = 21) received equal amounts of vehicle (saline). At 2 weeks and 5 weeks after fracture, the femora were analyzed by means of biomechanics, µ-CT, histology, and immunohistochemistry. In addition, Western blot analysis for the quantification of protein expression was performed at 2 weeks after fracture.

**Figure 2 ijms-25-00755-f002:**
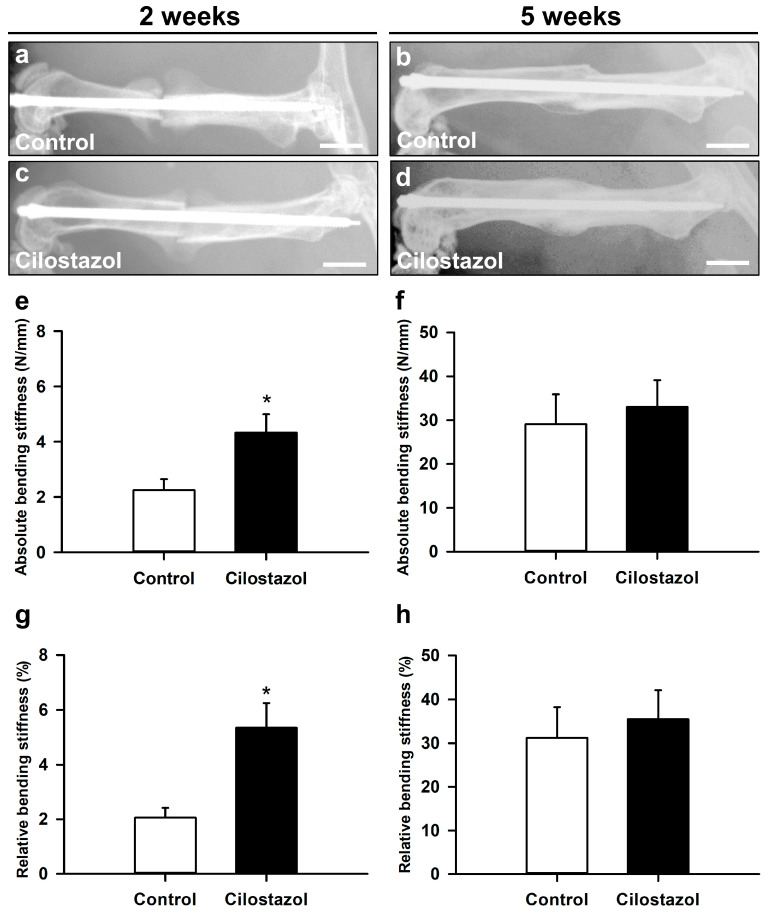
(**a**–**d**) Representative X-rays of fractured mouse femora stabilized by an intramedullary screw in controls (**a**,**b**) and cilostazol-treated animals (**c**,**d**) at 2 (**a**,**c**) and 5 weeks (**b**,**d**) after fracture. Scale bars: 2 mm. (**e**–**h**) Biomechanical analysis of absolute (**e**,**f**) and relative (**g**,**h**) bending stiffness in controls (white bars) and cilostazol-treated animals (black bars) at 2 ((**e**,**g**); *n* = 8) and 5 weeks ((**f**,**h**); *n* = 9) after fracture. Data are given in absolute values (N/mm) and in percent of the contralateral, non-fractured femora (%). Mean ± SEM; * *p* < 0.05 vs. control. All parametric data analyses were performed using unpaired Student’s *t*-test.

**Figure 3 ijms-25-00755-f003:**
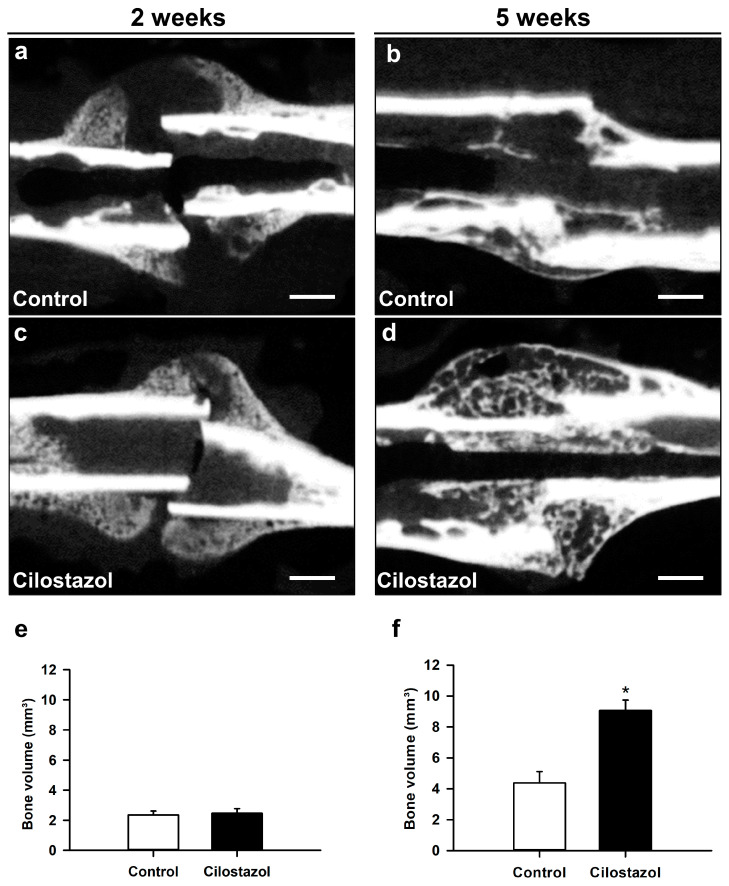
(**a**–**d**) Representative µCT images of mouse femora in controls (**a**,**b**) and cilostazol-treated animals (**c**,**d**) at 2 (**a**,**c**) and 5 weeks (**b**,**d**) after fracture. Scale bars: 0.75 mm. (**e–h**) µCT analysis of bone volume (**e**,**f**) and trabecular separation (**g**,**h**) in controls (white bars) and cilostazol-treated animals (black bars) at 2 ((**e**,**g**); *n* = 8) and 5 weeks ((**f**,**h**); *n* = 9) after fracture. Mean ± SEM; * *p* < 0.05 vs. control. (**e**,**h**) Non-parametric data; analysis performed by Mann–Whitney U-test. (**f**,**g**) Parametric data; analysis performed by unpaired Student’s *t*-test.

**Figure 4 ijms-25-00755-f004:**
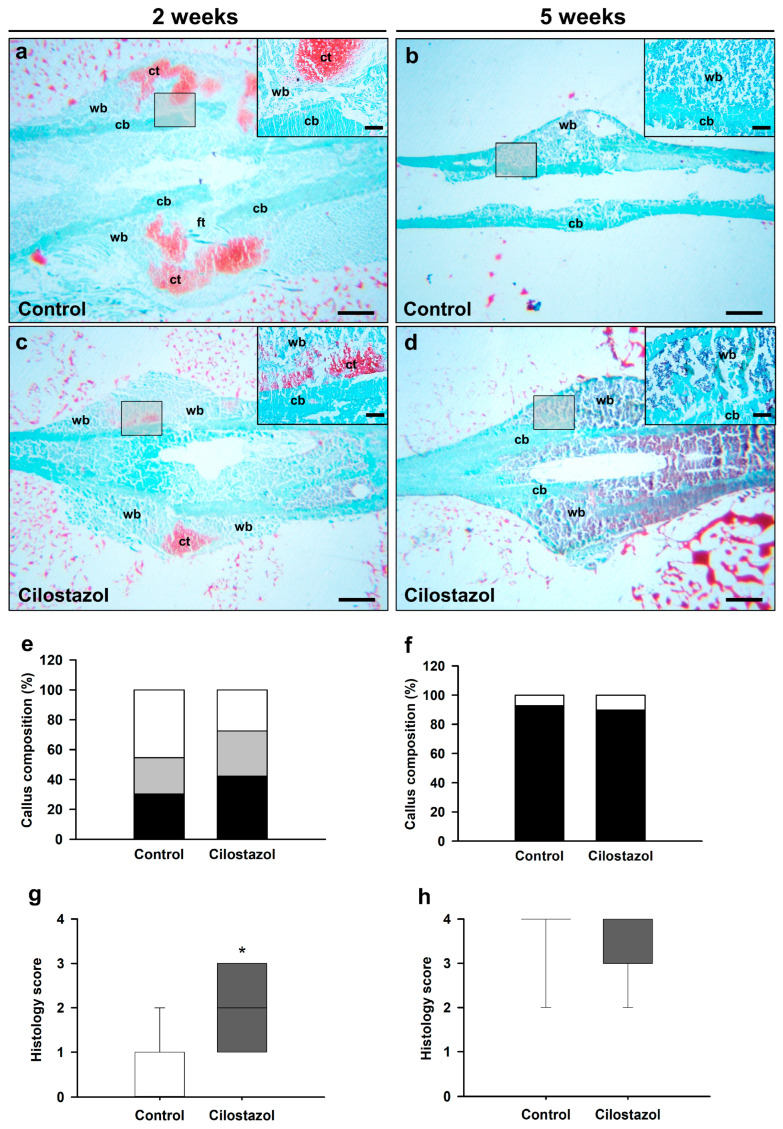
(**a**–**d**) Representative histological images of safranin-O-stained femora of controls (**a**,**b**) and cilostazol-treated animals (**c**,**d**) at 2 (**a**,**c**) and 5 weeks (**b**,**d**) after fracture. Scale bars: 0.75 mm. A higher magnification of the callus tissue is provided for better visualization of fibrous tissue (ft), cartilaginous tissue (ct), woven bone (wb), and cortical bone (cb). Scale bars 0.12 mm. (**e**,**f**) Callus composition (%), including fibrous tissue (white), cartilaginous tissue (gray), and osseous tissue (black), of femora of controls (*n* = 8–9) and cilostazol-treated animals (*n* = 8–9) at 2 (**e**) and 5 weeks (**f**) after fracture. (**g**,**h**) Analysis of the quality of fracture bridging by the histology score of femora of controls (*n* = 8–9) and cilostazol-treated animals (*n* = 8–9) at 2 (**g**) and 5 weeks (**h**) after fracture. Data are given in median and quartiles; * *p* < 0.05 vs. control. (**e**) Osseous and cartilaginous tissue data are parametric; analysis performed by unpaired Student’s *t*-test. (**e**) Fibrous tissue data (**f**–**h**) are non-parametric; analysis performed by Mann–Whitney U-test.

**Figure 5 ijms-25-00755-f005:**
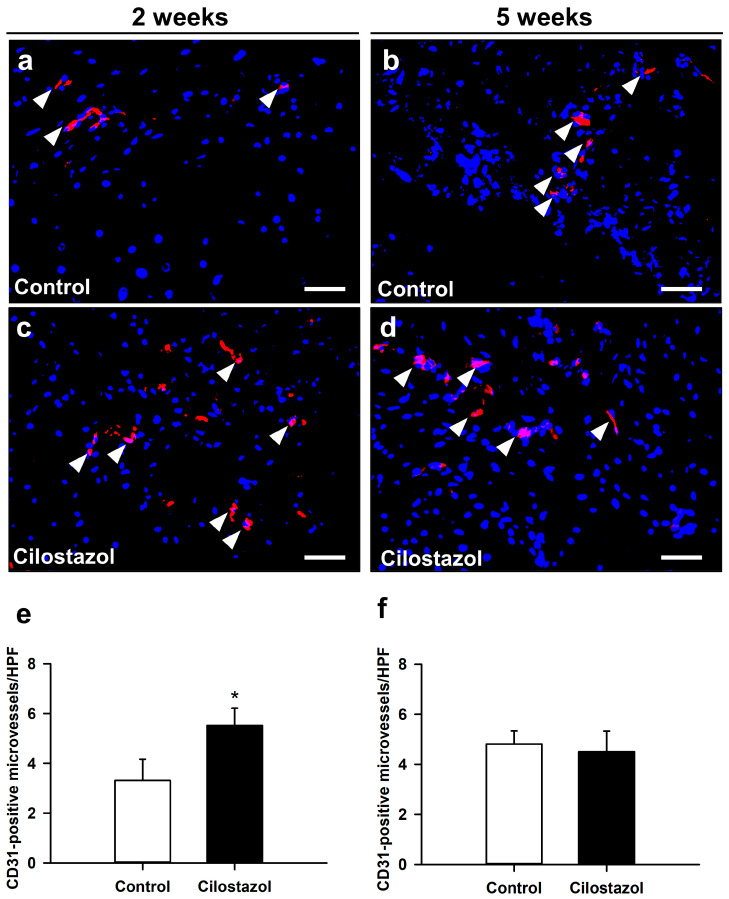
(**a–d**) Representative immunohistochemical images of CD31-positive microvessels (arrowheads) within the callus tissue of controls (**a**,**b**) and cilostazol-treated animals (**c**,**d**) at 2 weeks (**a**,**c**) and 5 weeks (**b**,**d**) after fracture. Scale bars: 50 µm. (**e**,**f**) CD31-positive microvessels/HPF in controls (white bars) and cilostazol-treated mice (black bars) at 2 ((**e**); *n* = 8) and 5 weeks ((**f**); *n* = 9) after fracture. Mean ± SEM; * *p* < 0.05 vs. control. All data are parametric; analysis performed by unpaired Student’s *t*-test.

**Figure 6 ijms-25-00755-f006:**
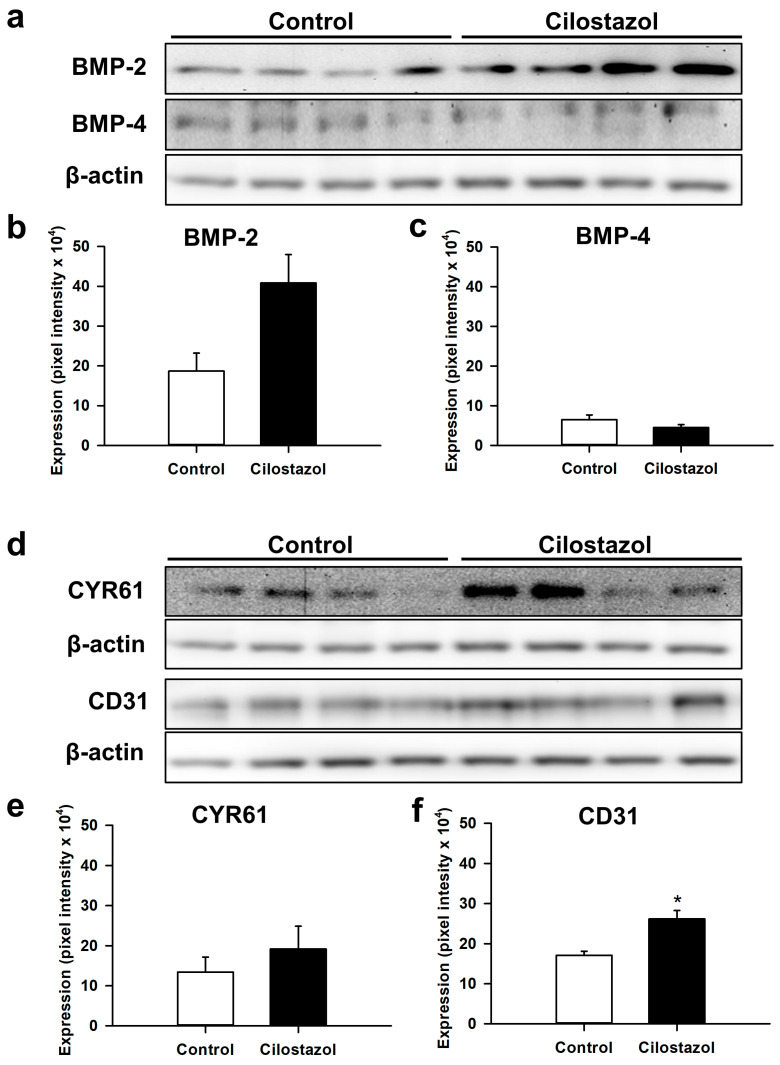
(**a**) Representative Western blots of BMP-2 and BMP-4 expression within the callus tissue of controls and cilostazol-treated mice at 2 weeks after fracture. (**b**,**c**) Expression of BMP-2 (**b**) and BMP-4 (**c**) within the callus tissue of controls (white bars, *n* = 4) and cilostazol-treated mice (black bars, *n* = 4) at 2 weeks after fracture. Mean ± SEM. (**d**) Representative Western blots of CYR61 and CD31 expression within the callus tissue of controls and cilostazol-treated mice at 2 weeks after fracture. (**e**,**f**) Expression of CYR61 (**e**) and CD31 (**f**) within the callus tissue of controls (white bars, *n* = 4) and cilostazol-treated mice (black bars, *n* = 4) at 2 weeks after fracture. Mean ± SEM; * *p* < 0.05 vs. control. (**g**) Representative Western blots of PI3K and RUNX2 expression within the callus tissue of controls and cilostazol-treated mice at 2 weeks after fracture. (**h**,**i**) Expression of PI3K (**h**) and RUNX2 (**i**) within the callus tissue of controls (white bars, *n* = 4) and cilostazol-treated mice (black bars, *n* = 4) at 2 weeks after fracture. Mean ± SEM; * *p* < 0.05 vs. control. (**b**,**f**) Non-parametric data; analysis performed by Mann–Whitney U-test. (**c**,**e**,**h**,**i**) Parametric data; analysis performed by unpaired Student’s *t*-test.

**Table 1 ijms-25-00755-t001:** Trabecular thickness (mm) and trabecular number (1/mm) in controls and cilostazol-treated mice at 2 (*n* = 8) and 5 weeks (*n* = 9) after fracture. Mean ± SEM. The trabecular thickness data at 2 weeks are non-parametric; analysis performed by Mann–Whitney U-test. Trabecular thickness at 5 weeks and trabecular number at 2 and 5 weeks are parametric data; analysis performed by unpaired Student’s *t*-test.

	2 Weeks	5 Weeks
Trabecular thickness (mm)		
Control	0.12 ± 6.53 × 10^−3^	0.18 ± 14.67 × 10^−3^
Cilostazol	0.11 ± 8.84 × 10^−3^	0.21 ± 6.8 × 10^−3^
Trabecular number (1/mm)		
Control	2.34 ± 0.25	2.38 ± 0.10
Cilostazol	1.8 ± 0.18	2.84 ± 0.21

## Data Availability

Data are contained within the article.

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
