# Peer review of "Cilostazol Stimulates Angiogenesis and Accelerates Fracture Healing in Aged Male and Female Mice by Increasing the Expression of PI3K and RUNX2"

_ijms, 2024, doi:10.3390/ijms25020755_

Round 1

Reviewer 1 Report

Comments and Suggestions for Authors

Paper titled (Cilostazol Accelerates Fracture Healing in Aged Nice by Increasing the Expression of PI3K and RUNX2) by Menger et al. demonstrated the effect of the antioxidant, cilostazol, in accelerating healing of experimental fractures in aged mice and claimed this effect was mediated throught affecting the expression of PI3K and RUNX2. The rational of the experiment is fine but the excustion suffers from defaults & experiemntal design does not include appropriate controls in addition to using singld does of Calycosin makes the results weak and not reliable. SOme statistical analysis are not correctly done & the whole paper requires extensive revision to reach the level of acceptance. Also why WB was done at week 2 and not week 5? This is not logic!!  Kindly find the reasons below:

1- Title: did not show what was the source of fracture.-  Title: correct "Nice" to "Mice"

2- Mention in the title if used male or female mice

3- Angiogenesis is mentioned in key words and intro but not highlighted in title

4- Abstract: should be amended by some numerical values

5- Introduction line 53: no need for liver and kidney

6-Line 56: authors wrote "Therefore, we analyzed" it does not make sense, please use a better verb.

7- key words: needs more specific items about the model

8- in 2.1.: authors did not mention the housing conditions & number of animals per group ...etc & how they minimized animal suffering. These are important details

9- Why authors dissolved cilostazol in saline, NOT in dist water for oral use.

10- A graphic illustartion for the study design and periodical steps will be valuable

11- I find the study design is not symmetric, some parameters measured at week 2 and week 5 but others (WB) measured only at week 2. This was we acnnot take a strong conclusion from the study

12- The number of animals in each group is NOT symmetric 

13- In methods: describe more the structure indices

14- The method of softening bone tissue should be clearly described 

15- I believe one sham control group is missing in this study.

16- Group nomenclature is not accurate (group received the vehicle only is better nominated a "vehicle control group" and another is a disease control..etc

17- Steps of antigen retrieval protocol and immunohistochemistry should be given

18- In 2.8.: "The number of CD31-positive microvessels 158 within each HPF was counted" may be not accurate and better to measure the whole area of immunostaining

19- In results: histo score should be histology score

20- Histology scores are ordinal data cannot be represented as mean & SE
should be median & quartiles. 

21- Why authors measure total PI3K Not the phosphorylated form?

22-   The method and software used for quantification in WB analysis should be added.

23-     Use appropriate abbreviations for minutes, seconds...etc

24-     Mention "n" in each illustation individually

25- In each illustration mention the type of the presented data & the statistical test applied for analysis 

Comments on the Quality of English Language

minor corrections needed

Author Response

Review of the manuscript "Cilostazol accelerates fracture healing in aged mice by increasing the expression of PI3K and RUNX2"

We appreciate the fair and constructive comments of the reviewers. In the following, please find our point-by-point reply.

Reviewer 1

Paper titled (Cilostazol Accelerates Fracture Healing in Aged Nice by Increasing the Expression of PI3K and RUNX2) by Menger et al. demonstrated the effect of the antioxidant, cilostazol, in accelerating healing of experimental fractures in aged mice and claimed this effect was mediated throught affecting the expression of PI3K and RUNX2. The rational of the experiment is fine but the excustion suffers from defaults & experiemntal design does not include appropriate controls in addition to using single does of Calycosin makes the results weak and not reliable. Some statistical analysis are not correctly done & the whole paper requires extensive revision to reach the level of acceptance. Also why WB was done at week 2 and not week 5? This is not logic!!  Kindly find the reasons below:

1- Title: did not show what was the source of fracture.-  Title: correct "Nice" to "Mice"

Reply: According to the comment of the reviewer we corrected the error “nice” to “mice. Although the source of the fracture is not mentioned in the title, the fracture model is described in detail in the material and methods section of the manuscript.

2- Mention in the title if used male or female mice

Reply: In the present study we used male and female mice to achieve an appropriate number of animals, due to the long breeding time of aged mice (16-18 months). Notably, male and female mice were equally distributed in both study groups. This information is now given in title and the material and method section the revised version of the manuscript.

(See revised manuscript; page 12; line 267; marked in yellow)

According to the comment of the reviewer now mention the term male and female.

3- Angiogenesis is mentioned in key words and intro but not highlighted in title

Reply: According to the comment of the reviewer we now emphasize the effect of cilostazol on angiogenesis in the title of manuscript.

4- Abstract: should be amended by some numerical values

Reply: According to the comment of the reviewer we now added numerical values within the abstract of the manuscript.

(See revised manuscript; page 1; lines 24-29; marked in yellow)

5- Introduction line 53: no need for liver and kidney

Reply: According to the comment of the reviewer we deleted the phrases liver and kidney.

6-Line 56: authors wrote "Therefore, we analyzed" it does not make sense, please use a better verb.

Reply: According to the comment of the reviewer we changed the wording to “Accordingly, we analyzed in the present study for the first time…”

(See revised manuscript; page 2; line 59; marked in yellow)

7- key words: needs more specific items about the model

Reply: According to the comment of the reviewer we added the keywords “femur” and “screw” I the key word.

(See revised manuscript; page 1; line 33; marked in yellow)

8- in 2.1.: authors did not mention the housing conditions & number of animals per group ...etc & how they minimized animal suffering. These are important details

Reply: According to the comment of the reviewer we added housing conditions within the revised version of the manuscript. The analgesia procedures, to minimize animals suffering are already mentioned in paragraph 4.3 in the original version of the manuscript. 

(See revised manuscript; 12 and 13; lines 270-273 and lines 309-313; marked in yellow)

9- Why authors dissolved cilostazol in saline, NOT in dist water for oral use.

Reply: We found that the cilostazol tablets dissolve well in saline. Hence, we used it in the present study.

10- A graphic illustration for the study design and periodical steps will be valuable

Reply: According to the comment of the reviewer we now provide a graphic illustration of the study design and periodical steps.

(See revised manuscript; Figure 1, page 2; lines 71-78)

11- I find the study design is not symmetric, some parameters measured at week 2 and week 5 but others (WB) measured only at week 2. This was we cannot take a strong conclusion from the study

Reply: The Western blot analysis was done only at 2 weeks after fracture for two reasons:

    • Growth factor expression is predominant at 2 weeks after bone injury, when the mineralized callus starts forming and cartilaginous tissue is replaced by woven bone. [1]
    • Moreover, at 5 weeks after fracture there is not sufficient callus tissue for harvesting, due to the advanced fracture consolidation.
  1. T. Histing, D. Stenger, C. Scheuer, W. Metzger, P. Garcia, J.H. Holstein, M. Klein, T. Pohlemann, and M.D. Menger, Pantoprazole, a proton pump inhibitor, delays fracture healing in mice, Calcif Tissue Int. 90 (2012) 507-14. http://www.ncbi.nlm.nih.gov/pubmed/22527206.

12- The number of animals in each group is NOT symmetric 

Reply: The reviewer is right that the number of animals between the different observation time points is not symmetrical (n = 8 at 2 weeks; n = 9 at 5 weeks). However, the number of animals observed between the two study groups is symmetric.

13- In methods: describe more the structure indices

Reply: According to the comment of the reviewer we now provide a more detailed description of the structural indices calculated in our histomorphometric analysis, which reads as follows:

The total callus area is defined as the total area of bone, cartilaginous and fibrous callus area. The bone callus area usually originates at 2 weeks after fracture from the cortical bone on the outside of the callus tissue. The cartilaginous callus is clearly indicated by the red color by Safranin-O staining. The fibrous callus is characterized by long green fibers within the center of the callus tissue.

(See revised manuscript; page 14; lines 351-355; marked in yellow)

14- The method of softening bone tissue should be clearly described 

Reply: In the present study we used the Kawamoto film method, which relies on cryopreservation of the histological sections. Hence, the bone tissue was not softened. This method provides a great quality of immunohistochemical staining, such as CD31 [1].

[1] T. Kawamoto and K. Kawamoto, Preparation of thin frozen sections from nonfixed and undecalcified hard tissues using Kawamot's film method (2012), Methods Mol Biol. 1130 (2014) 149-164. https://www.ncbi.nlm.nih.gov/pubmed/24482171.

(See revised manuscript; page 14; line 345; marked in yellow)

15- I believe one sham control group is missing in this study.

Reply: In the present study two experimental study groups where compared:

Control: receiving daily 100µL of vehicle (saline) by oral gavage.

Cilostazol: receiving daily 30 mg/kg body weight cilostazol dissolved in 100 µL saline by oral gavage.

In both study groups geriatric mice with the age of 16-18 months were used. Hence, we do not think that there is a sham group missing in the present study. To clarify that the vehicle, which was received by the control animals was saline, we changed the wording in the revised version of the manuscript.

(See revised manuscript; page 12; line 282; marked in yellow)

16- Group nomenclature is not accurate (group received the vehicle only is better nominated a "vehicle control group" and another is a disease control..etc

Reply: According to the reply of comment 15, we feel that the group nomenclature (Control and Cilostazol) is correct in the present study.

17- Steps of antigen retrieval protocol and immunohistochemistry should be given

Reply: According to the comment of the reviewer we now provide a more detailed protocol for the immunohistochemistry, which reads as follows:

For immunohistochemistry the sections were stored für 60 secs in 100% ethanol. After washing in phosphate buffered saline (PBS), the sections were permeabilization in PBS with 0.2 % Triton-X (PBST) for 10 mins. The process was blocked by 5 % goatserum in PBST für 1 h. For the immunohistochemical detection of microvessels, sections were stained with a monoclonal rat anti-mouse antibody against the endothelial cell marker CD31 (1:100; Abcam, Cambridge, UK) overnight in PBST with 5% goatserum. Subsequently, the sections were washed again in PBS, and stained with a goat anti-rat IgG-Alexa555 antibody as secondary antibody (1:100; Life Technol-ogy, Eugene, USA) in PBS für 1 h. Cell nuclei were stained with Hoechst 33342 (2 µg/mL; Sigma-Aldrich, Taufkirchen Germany). Finally, the sections were covered with glycerin gelatine.

(See revised manuscript; page 14; lines 363-373; marked in yellow)

18- In 2.8.: "The number of CD31-positive microvessels within each HPF was counted" may be not accurate and better to measure the whole area of immunostaining

Reply: In the present study we used equally distributed HPFs to evaluate the number of CD31-positive microvessels. In 2-week specimens, one HPF (×400 magnification) was placed in a standardized manner in the central region of the callus (former fracture gap), while five additional HPFs were placed at each site within the periosteal region of the callus. Due of the reduced size of the callus at 5 weeks, only 3 additional HPFs were placed at each site within the periosteal region of the callus. This standardized distribution of the HPFs allowed a reliable analysis of the number of CD-31 positive microvessels:

2 weeks:

5 weeks:

The reviewer is right that also the whole callus area may be examined, however this would be only possible with a lower magnification. The x400 magnification of the HPFs in the present study allowed a reliable manual counting of the CD31-positive microvessels, which we feel would not be possible with a lower magnification.

19- In results: histo score should be histology score

Reply: According to the comment of the reviewer we changed the wording from histo score to histology score.

(See revised manuscript; page 14; line 357; marked in yellow)

20- Histology scores are ordinal data cannot be represented as mean & SE
should be median & quartiles. 

Reply: According to the comment of the reviewer now present the histology score with median and quartiles in the revised version of the manuscript.

(See revised manuscript; Figure 4)

21- Why authors measure total PI3K Not the phosphorylated form?

Reply: We agree with the author that also the phosphorylated form of PI3K may be of interest. However, the literature we found and discussed on the effects of PI3K on bone regeneration only mentions the not phosphorylated form of PI3K. This includes the beneficial effects of PI3K on the proliferation of osteoblasts and periosteal cells and the possible stimulation of bone regeneration [1, 2]. Moreover, the current literature demonstrating a cilostazol-induced increase of PI3K expression, resulting in tissue-protective and pro-angiogenic effects, only analyzes the non-phosphorylated form of PI3K [3, 4]. Hence, we analyzed the effects of cilostazol treatment on the expression of the non-phosphorylated form of PI3K within the callus.

  1. Guntur, A.R.; Rosen, C.J. The skeleton: a multi-functional complex organ. New insights into osteoblasts and their role in bone formation: the central role of PI3Kinase.  Endocrinol.2011211, 123–130. https://doi.org/10.1530/joe-11-0175.
  2. Scanlon, V.; Walia, B.; Yu, J.; Hansen, M.; Drissi, H.; Maye, P.; Sanjay, A. Loss of Cbl-PI3K interaction modulates the periosteal response to fracture by enhancing osteogenic commitment and differentiation. Bone201795, 124–135. https://doi.org/10.1016/j.bone.2016.11.020.
  3. Mohamed, M.Z.; Hafez, H.M.; Zenhom, N.M.; Mohammed, H.H. Cilostazol alleviates streptozotocin-induced testicular injury in rats via PI3K/Akt pathway. Life Sci.2018198, 136–142. https://doi.org/10.1016/j.lfs.2018.02.038.
  4. Shi, M.-Q.; Su, F.-F.; Xu, X.; Liu, X.-T.; Wang, H.-T.; Zhang, W.; Li, X.; Lian, C.; Zheng, Q.-S.; Feng, Z.-C. Cilostazol suppresses angiotensin II-induced apoptosis in endothelial cells.  Med. Rep.201613, 2597–2605. https://doi.org/10.3892/mmr.2016.4881.

22-   The method and software used for quantification in WB analysis should be added.

Reply: According to the comment of the reviewer we now describe the method and software used for the quantification of Western blot analysis, which reads as follows:

For quantification of protein expression, the Lagimage 1D software (Kapelan Bio-Imaging GmbH, Leipzig, Germany) was used. The software automatically detects the position of the Western blot bands and quantifies their pixel intensity.

(See revised manuscript; page 15; line 398-400; marked in yellow)

23-     Use appropriate abbreviations for minutes, seconds...etc

Reply: According to the comment of the reviewer we now use appropriate abbreviations for minutes and seconds in the revised version of the manuscript.

24-     Mention "n" in each illustation individually

Reply: According to the comment of the reviewer now mention “n” for each illustration within the figure.

25- In each illustration mention the type of the presented data & the statistical test applied for analysis 

 Reply: According to the comment of the reviewer we now mention in each illustration if data is parametric or non-parametric as well as the statistical analysis.

Reviewer 2 Report

Comments and Suggestions for Authors

Menger et al studied the effect of Cilostazol on fracture healing in aged mice.  After careful evaluation of the manuscript, I found some key issues that need to be addressed:

1.      In the title there should be mice instead of nice.

2.      In X-ray (Figure 1) and microCT (Figure 2) at 2 months there is no visible bone bridging. How can it increase the stiffness at 2 months in cilostazol-treated mice?  

3.      I suggest the author show other trabecular parameters such as Tb.N and Tb.Th.

Author Response

Review of the manuscript "Cilostazol accelerates fracture healing in aged mice by increasing the expression of PI3K and RUNX2"

We appreciate the fair and constructive comments of the reviewers. In the following, please find our point-by-point reply.

Reviewer 2

  1. In the title there should be mice instead of nice.

Reply: According to the comment of the reviewer we corrected this error in the revised version of the manuscript.

  1. In X-ray (Figure 1) and microCT (Figure 2) at 2 months there is no visible bone bridging. How can it increase the stiffness at 2 months in cilostazol-treated mice?

Reply: The reviewer is correct that at 2 weeks after fracture bone bridging is not yet visible in most controls and cilostazol-treated mice. However, our histological analysis demonstrates a significantly increased histology score at 2 weeks after fracture indicating that in cilostazol-treated mice, the fracture bridging is accelerated when compared to controls. In the histology score both cortices were analyzed for bone bridging (2 points), cartilage bridging (1 point) or bridging with fibrous tissue (0 point). This scoring system results in a maximum of 4 points for each specimen, indicating complete bone bridging. Accordingly, the cortices of cilostazol-treated animals are more likely bridged by cartilaginous or bone tissue when compared to controls, resulting in the observed improved bending stiffness at 2 weeks after fracture. 

  1. I suggest the author show other trabecular parameters such as Tb.N and Tb.Th.

Reply: According to the comment of the reviewer we added the trabecular number and thickness in a table within the revised version of the manuscript.

(See revised manuscript; Table 1; pages 4 and 6; lines 117-119; 126-131; marked in yellow)

Round 2

Reviewer 1 Report

Comments and Suggestions for Authors

The revised version of paper titled (Cilostazol Stimulates Angiogenesis and Accelerates Fracture Healing in Aged Male and Female Mice by Increasing the Expression of PI3K and RUNX2) by Menger et al. is still of low quality and was very partly improved. Hence, I still find it hard to publish this paper in IJMS.

The study design lacks important control group (no fracture or sham) that must be added to the comparison and was neglected by the authors. 

Figure 4 : Safranin O staining is of low quality that is not accepted to be published 

Also the assay of biomarjkers by WB just after 2 weeks is not logic to find a signifcant change at this short period & it would be realistic to do them at the end of the study. 

Beside the previous comments

Comments on the Quality of English Language

FIne

Author Response

Review of the manuscript "Cilostazol accelerates fracture healing in aged mice by increasing the expression of PI3K and RUNX2"

We appreciate the fair and constructive comments of the reviewers. In the following, please find our point-by-point reply.

Reviewer 1

The revised version of paper titled (Cilostazol Stimulates Angiogenesis and Accelerates Fracture Healing in Aged Male and Female Mice by Increasing the Expression of PI3K and RUNX2) by Menger et al. is still of low quality and was very partly improved. Hence, I still find it hard to publish this paper in IJMS.

1- The study design lacks important control group (no fracture or sham) that must be added to the comparison and was neglected by the authors.

Reply: In the original version of the manuscript, we already provide data of the non-fractured contralateral femora. Of note, cilostazol treatment had no significant impact on the bending stiffness of the non-fractured femora 100.6 ± 5.0 N/mm (control) vs. 92.3 ± 5.2 N/mm (cilostazol)).

See revised version of the manuscript; page 3 lines 98-100; marked in yellow.

Moreover, an additional histological and µCT analysis of unfractured femora would not be expedient, since there is no callus tissue to analyze. Hence, a comparison to the fractured femora cannot be drawn.

Figure 4 : Safranin O staining is of low quality that is not accepted to be published

Reply: We improved the contrast setting in the provided images of Figure 4.

Also the assay of biomarkers by WB just after 2 weeks is not logic to find a significant change at this short period & it would be realistic to do them at the end of the study.

Reply: As stated in our previous response the expression of biomarkers and growth factors is predominant in the early phase of fracture healing. Moreover, due to the advanced stage of healing and remodeling there is not sufficient callus tissue to harvest for Western blot analysis at 5 weeks after surgery. Therefore, we feel that Western blot analysis only at 2 weeks after fracture is appropriate. Notably, this study design was also performed in various previous studies [1-8]

  1. T. Histing, D. Stenger, C. Scheuer, W. Metzger, P. Garcia, J.H. Holstein, M. Klein, T. Pohlemann, and M.D. Menger, Pantoprazole, a proton pump inhibitor, delays fracture healing in mice, Calcif Tissue Int. 90 (2012) 507-14. http://www.ncbi.nlm.nih.gov/pubmed/22527206.
  2. M.M. Menger, P. Bremer, C. Scheuer, M.F. Rollmann, B.J. Braun, S.C. Herath, M. Orth, T. Spater, T. Pohlemann, M.D. Menger, and T. Histing, Pantoprazole impairs fracture healing in aged mice, Sci Rep. 10 (2020) 22376. https://www.ncbi.nlm.nih.gov/pubmed/33361800.
  3. S.C. Herath, T. Lion, M. Klein, D. Stenger, C. Scheuer, J.H. Holstein, P. Morsdorf, M.F. Rollmann, T. Pohlemann, M.D. Menger, and T. Histing, Stimulation of angiogenesis by cilostazol accelerates fracture healing in mice, J Orthop Res. 33 (2015) 1880-7. http://www.ncbi.nlm.nih.gov/pubmed/26134894.
  4. T. Histing, K. Marciniak, C. Scheuer, P. Garcia, J.H. Holstein, M. Klein, R. Matthys, T. Pohlemann, and M.D. Menger, Sildenafil accelerates fracture healing in mice, J Orthop Res. 29 (2011) 867-73. http://www.ncbi.nlm.nih.gov/pubmed/21246617.
  5. M.M. Menger, B. Merscher, C. Scheuer, B.J. Braun, S.C. Herath, M.F. Rollmann, D. Stenger, T. Spater, T. Pohlemann, M.D. Menger, and T. Histing, Amlodipine accelerates bone healing in a stable closed femoral fracture model in mice, Eur Cell Mater. 41 (2021) 592-602. https://www.ncbi.nlm.nih.gov/pubmed/34027631.
  6. M.M. Menger, M. Stief, C. Scheuer, M.F. Rollmann, S.C. Herath, B.J. Braun, S. Ehnert, A.K. Nussler, M.D. Menger, M.W. Laschke, and T. Histing, Diclofenac, a NSAID, delays fracture healing in aged mice, Exp Gerontol. 178 (2023) 112201. https://www.ncbi.nlm.nih.gov/pubmed/37169100.
  7. T. Histing, C. Anton, C. Scheuer, P. Garcia, J.H. Holstein, M. Klein, R. Matthys, T. Pohlemann, and M.D. Menger, Melatonin impairs fracture healing by suppressing RANKL-mediated bone remodeling, J Surg Res. 173 (2012) 83-90. http://www.ncbi.nlm.nih.gov/pubmed/20888595.
  8. T. Histing, S. Kuntz, D. Stenger, C. Scheuer, P. Garcia, J.H. Holstein, M. Klein, T. Pohlemann, and M.D. Menger, Delayed fracture healing in aged senescence-accelerated P6 mice, J Invest Surg. 26 (2013) 30-5. http://www.ncbi.nlm.nih.gov/pubmed/23273143.

Round 3

Reviewer 1 Report

Comments and Suggestions for Authors

Dear authors, I am sorry, I cannot find your reply convincing to me,

Comments on the Quality of English Language

Fine

Author Response

According to the comments of the Editor we now present histological sections at a higher magnification.

We appreciate the fair and constructive comments of the academic editor. In the following, please find our point-by-point reply.

Academic editor

The paper is interesting, I verified that it contains the necessary controls, but a great problem is represented by the histology:

  1. a better section should be presented:

Reply: According to the comment of the academic editor we now provide a higher magnification of the callus tissue as inserts in revised figure 4. These images further show the morphological differences in fibrous, cartilaginous tissue and bone tissue.

(See revised version of the manuscript; figures 4a-d)

  1. the staining for bone cells should be performed, as after five week ossification should be evident.

Reply: The Safranin-O/fast green staining method provides histological sections with clearly visible red cartilaginous tissue. However, the staining also enables the visualization of bone tissue by the green color. The quantification of the callus composition is based on the protocol of Gerstenfeld et al. [1], who also used Safranin-O staining. Notably, cell nuclei of osteocytes are also visible in the now provided higher magnification images of the sections in revised figure 4, appearing in blue/green color.

(See revised version of the manuscript; figures 4a-d)

Reference:

  1. Gerstenfeld, L.C.; Wronski, T.J.; O Hollinger, J.; A Einhorn, T. Application of Histomorphometric Methods to the Study of Bone Repair. J. Bone Miner. Res. 2005, 20, 1715–1722. https://doi.org/10.1359/jbmr.050702